# Long-Term Monitoring Reveals Changes in the Small Mammal Community Composition and Co-Occurrence Patterns in the Diannan Area of Yunnan, China

**DOI:** 10.3390/biology14070897

**Published:** 2025-07-21

**Authors:** Jinyu Yang, Ting Jia, Wanlong Zhu, Xiaomi Yang

**Affiliations:** 1Key Laboratory of Ecological Adaptive Evolution and Conservation on Animals-Plants in Southwest Mountain Ecosystem of Yunnan Province Higher Institutes College, School of Life Sciences, Yunnan Normal University, Kunming 650500, China; 2Key Laboratory of Yunnan Province for Biomass Energy and Environment Biotechnology, Kunming 650500, China; 3Yunnan Key Laboratory of Integrated Traditional Chinese and Western Medicine for Chronic Disease in Prevention and Treatment, Yunnan University of Chinese Medicine, Kunming 650500, China

**Keywords:** small mammals, species diversity, co-occurrence network, community structure

## Abstract

Investigating small mammal diversity is critical for monitoring ecosystem health and assessing human impacts on biodiversity. In this study, we conducted a 13-year survey (2005–2017) in the Diannan area of Yunnan, China, analyzing species composition, diversity indices, and co-occurrence networks. We identified significant declines in species diversity and community complexity, driven primarily by climate factors (temperature and precipitation). Our findings highlight the urgent need for targeted conservation strategies to protect declining species and maintain habitat connectivity. This long-term dataset provides a scientific foundation for biodiversity management in mountainous ecosystems and demonstrates the value of integrated diversity and network analyses in conservation planning.

## 1. Introduction

The rapid expansion of human populations and associated activities, including intensive agricultural practices [1], industrial operations [2], and urban development [3,4], has precipitated severe ecosystem degradation, representing critical environmental challenges that demand urgent scientific attention and mitigation strategies [5]. Human activities have increased the concentration of greenhouse gases in the atmosphere and exacerbated global climate change [6], which has already had a significant impact on biodiversity and will persist in the future [7,8]. The structure and makeup of biological communities serve as crucial indicators of biodiversity. Continuous monitoring of communities is essential for mitigating the effects of human actions [9,10,11]. Monitoring over time also aids in evaluating the efficacy of conservation efforts. Analysis of long-term monitoring data reveals that targeted conservation measures, including the establishment of protected areas and ecological restoration initiatives, exert significant positive impacts on biodiversity metrics [12,13]. These empirical findings facilitate evidence-based optimization of conservation frameworks to achieve sustainable biodiversity management and ecosystem service provision [14,15].

Small mammals constitute a vital functional component of terrestrial ecosystems, facilitating nutrient cycling and energy flow through trophic interactions while mediating plant regeneration dynamics via seed dispersal services [16,17,18]. These organisms serve as an ideal group for investigation into the impact of environmental change on biodiversity due to their short generational cycles and heightened environmental sensitivity [19,20]. Their population dynamics directly mirror activity fluctuations of other species such as ungulates and birds [21,22]. By analyzing changes in the community structure and diversity parameters of small mammals, the integrity of forest–shrub habitats can be evaluated [23,24,25,26,27] and quantify anthropogenic disturbance effects on ecosystem health. Typically, the change is evaluated by calculating species richness, which is an important indicator of biodiversity [28]. However, even when species richness remains unchanged, community composition and the relative abundances of constituent species may also change over time. Therefore, when conducting surveys of small mammal communities, it is necessary to comprehensively consider the above variables [29]. Calculating the α diversity index is one of the common methods, such as the Shannon–Weiner diversity index [30,31].

In addition to the use of diversity indices to assess the community’s structural condition, changes in the relationships of species within a community can also be useful in predicting negative changes in biodiversity [32] and thus in assessing the impacts of climate and human activities on community structure [33,34,35]. Co-occurrence networks provide a usable theoretical framework for studying species relationships and the complexity of biological systems [36]. Co-occurrence networks consist of nodes and links, where species are nodes and their relationships are links. Core species in the community structure can be identified by analyzing the interspecific relationships, thereby implementing targeted protection [37]. Analyzing the temporal and spatial changes of co-occurrence networks has become an effective method to evaluate the impact of the environment on communities [38,39,40,41].

Therefore, this study aims to investigate the temporal variation characteristics of small mammal community structure and interspecific relationships in the Hengduan Mountain region, along with their climatic driving mechanisms. While previous studies have examined rodent and insectivore groups in this area [42,43,44], systematic understanding remains lacking regarding regional small mammal community composition, diversity dynamics, and their relationships with environmental factors. Accordingly, we selected the Diannan area of the Hengduan Mountains as our study area and conducted the first systematic analysis using 13 years of continuous monitoring data (2005–2017) on species composition, diversity indices, population abundance, and their correlations with environmental factors. By measuring species interactions represented by co-occurrence relationships, we aim to provide new insights into changes in small mammal communities. Furthermore, these long-term monitoring data provide a scientific basis for biodiversity conservation and ecosystem management in the Hengduan Mountain region.

## 2. Materials and Methods

### 2.1. Study Area

Located in the Dali Bai Autonomous Prefecture, Yunnan Province, the Diannan area covers 244 km^2^. It is positioned at 99°55′ E and 26°31′ N, boasting an average altitude of 2200 m, an annual mean temperature of 12.6 °C, and an average yearly precipitation of 762 mm. The terrain of Diannan area is complex, and its western and northern parts are mainly composed of mountains, and the vegetation here is dominated by Yunnan pine (*Pinus yunnanensis*). Beyond that, evergreen and deciduous broad-leaved forests are widespread in the hilly and basin areas. In the foothills, shrub plants such as shashanbo (*Vacciniun bracteatum Thunb*) and Dwarf thorn oak (*Quercus monimotricha*) are the main vegetation [45,46].

### 2.2. Field Methods

During the survey period, we established 48 sampling sites within the study area for sample collection (Figure 1). These sampling sites covered four typical habitats in the area, including agricultural fields, shrublands, residential areas, and vegetable fields around the residential areas.

Between 2005 and 2017, small mammal sampling was conducted at 48 sites in the Diannan region using standardized live-trapping protocols. Sherman live traps (200 × 120 × 100 mm^3^) were strategically deployed along small mammal runways and near burrow entrances, with a consistent 5–10 m inter-trap spacing to ensure uniform spatial coverage. Each sampling site maintained a constant trapping effort of 25 traps per night over three consecutive trapping nights. Fresh apple baits and insulating grass bedding were utilized to enhance capture success and ensure animal welfare. Trap monitoring occurred at 07:00–09:00 and 17:00–18:00 daily [47]. Captured individuals were processed following established protocols: recording capture metadata, measuring standard morphological parameters (body mass, head–body length, and tail length), and documenting diagnostic dental characteristics. Species identification was based on these morphological traits and current distribution records from the “Catalogue of Mammals of China”. All specimens received unique ear tags prior to release at capture locations to prevent recapture bias [48].

### 2.3. Climatic Data

Monthly historical climatic data for Diannan area (2005–2017) were obtained from the China Meteorological Data Service Center [http://data.cma.cn/ (accessed on 5 December 2023)]. The following 17 indicators were selected for subsequent analysis: temperature, total precipitation, skin temperature, relative humidity, evaporation, snowfall, PM2.5, PM10, surface ozone, surface carbon monoxide, surface sulfur dioxide, surface nitrogen dioxide, surface carbon dioxide, surface methane, wind speed, solar radiation, and sunshine duration.

### 2.4. Data Processing and Analysis

The species composition of small mammals was estimated as the relative contribution of each individual species in one year. The Berger–Parker dominance index (*C*′) was used to determine the dominant species, and when a species’ dominance index was ≥0.1, it was determined to be the dominant species [49], expressed by the following equation:(1)$$C^{\prime }=\Sigma (N_{i}/N)$$
where *N_i_* is the number of individuals in the species, and *N* is the total number of individuals in the community.

The biodiversity of small mammals was assessed using the Shannon–Wiener Diversity index (*H*′), the Pielou Index (*J*′), the Simpson dominance index (*D*), and the Margalef richness index (*D_m_*), expressed by the following equation:(2)$$H^{\prime }=-\sum_{j=1}^{S}P_{i}\ln P_{i}$$
where *H*′ is the Shannon–Wiener index for diversity, *S* is the total number of species, and *P_i_* is proportional to the total sample belonging to the species.

Evenness, representing the degree of equitability in the distribution of individuals across a set of species is(3)$$J^{\prime }=H^{\prime }/\ln(S)$$(4)$$D=1-{\sum_{j=1}^{S}\left(P_{i}\right)}^{2}$$(5)Dm=(S−1)/lnN 

We analyzed interannual differences in these four diversity indices with the Kruskal–Wallis test.

The species co-occurrence patterns for each year were analyzed by using the R software (version 4.10) co-occur package [50]; this method allows obtaining the significance of co-occurrence relationships for each species pair. The data obtained from the analysis were used to construct the co-occurrence network for different years; the observed co-occurrences of each species pair were used as edges and the species as nodes [37]. Species pairs with co-occurrence probabilities ≤0.05 were removed. Co-occurrence network graphs were plotted in Grephi software (version 0.10.1), and the co-occurrence network graph structure indices, like the average degree, density, modularity, and clustering coefficient, were calculated separately. Decadal trends of species richness, four diversity indices (the Simpson dominance index, the Shannon–Wiener diversity index, the Simpson dominance index, the Margalef richness index, and the Pielou evenness index), and four global structure indices of co-occurrence network (average degree, link density, modularity, and clustering coefficient) were analyzed by using generalized additive model (gam() function in the “mgcv” package) to capture their overall temporal trends [51,52].

Pearson correlation analysis and redundancy analysis (RDA) were employed to evaluate the relationship between abundance and climatic factors. Before conducting RDA, we performed a DCA (Detrended Correspondence Analysis), and the results showed that the longest axis was less than 4.00. This suggests that the relationship between climatic factors and the small mammal community of the Diannan area is relatively linear, so we chose RDA for further analysis. All of these analyses were conducted using the Vegan package in R (version 4.10).

## 3. Results

### 3.1. Species Collected During the Surveys

From year 2005 to 2017, a total of 22 species of small mammals belonging to four orders, seven families, and 15 genera were collected. Among them, 16 species belonging to three families of Rodentia were captured in the Diannan area (Table 1).

### 3.2. Variations in Species Composition

Among the small mammals captured in the Diannan area during the period 2005–2017, Rodentia captures were the largest, with an increasing trend, while captures of Eulipotyphla reached a maximum in 2016, with very few (<10) annual captures of Scandentia and Carnivora in the area, and no Carnivora were captured after 2013 (Figure 2).

Long-term monitoring data (2005–2017) revealed interannual variations in both abundance and community composition of small mammals in the Diannan area. *Apodemus chevrieri* maintained consistent ecological dominance throughout the study period, exhibiting peak absolute abundance in 2016 (*n* = 1271 individuals; Figure 3a) and maximum relative abundance in 2008 (67.55%). The *Eothenomys miletus* demonstrated stable population representation, maintaining >21.04% relative abundance in all years except 2008 (7.06%). Notably, these two species displayed significant positive population trends over the 13-year survey. On the contrary, the remaining 20 observed species exhibited declining relative abundance trends throughout the study period. The *Rattus norvegicus* periodically achieved dominant status during four distinct years (2005–2006 and 2010–2011) (Figure 3b, Appendix A). Generalized additive model (GAM) detected interannual increases in small mammals’ abundance (*p* < 0.001) (Figure 3a).

### 3.3. Variations in Species Diversity

The temporal dynamics of community diversity indices revealed interannual variations during the 2005–2017 monitoring period (Figure 4). All four indices exhibited a similar pattern of change: an initial decline (2005–2008), followed by a recovery (2009–2011), and then a subsequent decline (2012–2017). Specifically, the Shannon–Wiener index (*H*′) decreased by 21.4% (from 1.45 to 1.14) during 2005–2008, rebounded to its peak (1.48) in 2011, then declined to 1.16 by 2017. Similarly, the Margalef richness index (*D_m_*) plummeted 30.0% (from 1.60 to 1.12) through 2008, recovered to 1.56 by 2012, before declining to the lowest recorded value (1.11) in 2017. The Simpson dominance index (*D*) decreased by 22.9% (from 0.70 to 0.54) until 2008, peaking at 0.71 in 2011, then falling to its trough (0.53) in 2015. The Pielou evenness index (*J*′) reached its lowest recorded value (0.54) in 2008, recovering to the maximum observed value (0.81) in 2010, and subsequently declining through 2017.

The Kruskal–Wallis test confirmed significant differences in all four indices across years (*p* < 0.01), with post-hoc pairwise comparisons delineating specific interannual contrasts (Appendix A). Notably, an overall downward trend is evident across the Shannon–Wiener diversity, Margalef richness, Simpson dominance, and Pielou evenness indices (*p* < 0.001) (Figure 5a–e). This trend unfolded through a distinct pattern: All four indices initially underwent a rapid decline from 2005 to 2010, reached a turning point around 2011, signified by a rebound, but subsequently resumed their downward trajectory (Figure 5a–e).

### 3.4. Changes in Species Co-Occurrence and Networks

Analysis of species co-occurrence networks from 2005 to 2017 revealed distinct structural shifts (Figure 6). The number of network nodes (representing species) decreased progressively from 21 in 2005 to 14 in 2017. During this period, average degree (*p* = 0.042) and modularity declined significantly, while network density and the clustering coefficient increased (Figure 7).

Key extremal values emerged across the study years (Table 2): Average degree reached its maximum in 2005 (16.190) and minimum in 2010 (7.778); modularity peaked in 2006 (0.206) and troughed in 2008 (0.015); graph density ranged from 0.660 (2007) to 0.983 (2016); and the clustering coefficient varied between 0.863 (2006) and 0.983 (2016).

On average, more than 66.71% of species pairs remained associated in the following year, with about one-third of them losing their links (33.29% on average). The new co-occurrences gained in the following year accounted for 29.37% of the total on average (Table 3).

### 3.5. Association Patterns Between Climate Factors and Small Mammal Abundance

Pearson correlation analyses (Figure 8) between the abundance of small mammals and climatic factors revealed positive correlations between abundance versus temperature, total precipitation, skin temperature, relative humidity, surface carbon dioxide, and surface methane, and a negative correlation between abundance and the remaining climatic factors.

The first sorting axis explained 34.04% of the variance, while the second axis accounted for an additional 4.83%. Together, the first two axes cumulatively explained 38.87% of the variance in the relationship between the population size and climatic factors. Moreover, the results suggested that temperature, total precipitation, relative humidity, surface sulfur dioxide, surface carbon dioxide, surface ozone, and sunshine duration are the main climatic factors influencing the structure of the Diannan small mammals community (Figure 9).

## 4. Discussion

Over 13 years of continuous monitoring (2005–2017), 22 species of small mammals were recorded in the Diannan area, with significant interannual fluctuations in species richness. Compared with other regions in the Hengduan Mountains, the number of small mammal species in the Diannan area was higher than that in the Baicaoling Mt. (21 species) [53] but lower than the Three Parallel Rivers Area (44 species) [54] and the Wuliang Mountain Area (26 species) [55]. The peak diversity occurred in 2005 (21 species), while the lowest record was observed in 2010 (nine species), indicating pronounced ecosystem dynamism. This was similar to the results of long-term monitoring of small mammals in other regions [15,56]. *Eothenomys miletus* and *Apodemus chevrieri* have become dominant species, exhibiting relative abundance increases of 9.78% and 3.84%, respectively. In contrast, the remaining 20 species showed declining relative abundance trends, with all populations demonstrating random interannual fluctuations.

Notably, *Dremomys pernyi*, *Rupestes forresti*, *Callosciurus erythraeus*, and *Mustela sibirica* were not captured for over three consecutive years. Although community composition biases may arise from species-specific responses to trapping devices or bait preferences, we preliminarily propose that the persistent absence of these species suggests potential population threats and elevates their risk of local extinction. To test this hypothesis, subsequent investigations will implement multi-type trapping strategies (e.g., various Sherman trap models) combined with diversified baits for supplementary sampling in the study area.

A more favorable biological environment typically enhances species diversity through increased species richness, higher evenness, and reduced dominance [57,58]. Consistent with our hypothesis, the Shannon–Weiner index, Margalef richness index, and Gini–Simpson index exhibited significant interannual declines. However, species richness increased substantially during the study period. Critically, we attribute this apparent paradox to the dramatic population growth of two dominant species (*Eothenomys miletus* and *Apodemus chevrieri*), which has directly driven the decline of regional small mammal diversity. This phenomenon stems from intensified interspecific competition and disproportionate monopolization of key resources, ultimately suppressing other species’ viability within the ecosystem.

Our results indicated that the number of associated species of small mammals in the area has changed over time. The loss of one-third of species links indicates that environmentally constrained species relationships are increasing, which may be due to the reduction in environmental heterogeneity [59]. Moreover, the number of species positively associated with other species showed a downward trend. This might be linked to the contraction of local habitat area and food resources, compelling species that formerly inhabited disparate regions to converge in habitats like farmlands replete with more abundant food resources, which might warrant further attention. Increases in positively associated species over time potentially heighten competition for space and resources, especially among related taxa sharing similar functional morphology or feeding strategies [60].

To eliminate the contingency brought by extremely rare species and enhance the efficiency and accuracy of the co-occurrence pattern analysis, we have removed the species pairs with a co-occurrence probability lower than 0.05. However, excluding extremely rare species might also lead to the neglect of potentially important ecological connections between them and other species. The only species pairs with co-occurrence probabilities consistently greater than 0.05 over the 13-year period were the six species pairs formed by the four species of *E. miletus*, *A. chevrieri*, *R. norvegicus,* and *R. tanezumi*, suggesting that co-occurrence between the four is more stable. It is worth noting that the stable co-occurrence relationship between house mice and wild mice suggests that, at least at the sampling sites, there is a gradual overlap of ecological niches between house mice and wild mice. However, including the four species mentioned above, all 22 species captured here in the Diannan area showed declines in co-occurrence relationships, most of which were due to declines in the frequency of occurrence of the species. Species that undergo a significant decrease in both their individual abundances and the positive correlations with other species in co-occurrence are more likely to face the risk of local extinction themselves [61] and tend to have a negative impact on related species.

The structure metrics of the co-occurrence network structure metrics generally reflect the community’s robustness as well as resistance to extinction [62,63]. Therefore, a comprehensive analysis combining changes in diversity indices and co-occurrence networks can help to more comprehensively determine the future direction of the species community [37,64]. Using the data from the years 2005 and 2017 as an example, the diversity index is lower and significantly different in 2017 than in 2005, while at the same time, the co-occurrence network appears to have a decrease in average degree and modularity and an increase in graph density and clustering coefficient. It is hypothesized that this phenomenon may be related to the fact that the increase in the area of human land use, such as housing, has drastically reduced the area of animal habitat and that the frequency of occurrence of species such as weasels at the site has declined significantly, resulting in an increase in the frequency of interspecific communication and an intensification of interspecific competition, even though fewer species are included in the network.

A decrease or disappearance of a species in the area would have a negative impact on most small mammals in that area [65]. Meanwhile, in 2005 and 2006, for example, there was no significant difference in species diversity between the years because of the temporal proximity, but there was still a difference in the co-occurrence network structure indicators between the two years, which suggests that the pattern of co-occurrence between species may also change significantly in areas where species diversity has stabilized in a short period of time. Our findings corroborate that the species co-occurrence is a dynamic pattern [35].

The distribution of small mammals is closely related to the environment [19]. Investigations show that the abundance of small mammals in the Diannan area is mainly influenced by temperature, sunshine duration, and precipitation. Optimal temperatures facilitate stable metabolic responses in animals [66] and shorten the lifespan of small mammals [67], thereby influencing population dynamics. Precipitation indirectly affects small mammal abundance by altering plant productivity within habitats [68], while populations simultaneously respond to rainfall variations through shifts in spatial distribution and demographic changes [27]. Sunshine duration may indirectly impact populations by modifying animal activity patterns and physiological traits [69]. Changes in greenhouse gas concentrations can affect the structure and abundance of plant communities, thereby indirectly exerting adverse effects on the physiological processes of small mammals [70]. Climate change during the investigation period was also one of the factors contributing to the changes in the community structure of small mammals in this area.

Long-term animal monitoring facilitates ecological restoration by providing critical baseline data for deciphering community dynamics [71]. Most previous studies on small mammal community structure have primarily focused on temporal and spatial variations in diversity and relative abundance [15,72,73,74]. Integrating species diversity analysis with co-occurrence networks not only enables a more comprehensive understanding of community succession trajectories [65] but also offers scientific foundations for habitat quality assessment and conservation practices. By combining species composition analysis with co-occurrence network pattern analysis, our study revealed significant structural shifts in the small mammal community of the Diannan area, manifested through species loss, reduced diversity levels, and simplified interaction networks. These findings provide crucial insights for developing targeted biodiversity conservation strategies in the region.

## 5. Conclusions

In this study, we documented the community dynamics of 22 small mammal species (four orders, seven families, and 15 genera) in the Diannan area, Yunnan Province, through a 13-year survey (2005–2017), revealing *Eothenomys miletus* and *Apodemus chevrieri* as dominant species. We observed significant declines in diversity indices (Shannon–Wiener, Margalef, Simpson, and Pielou) and a simplification of co-occurrence networks, with species nodes decreasing from 21 to 14. Our analyses identified temperature, precipitation, and sunshine duration as key climatic drivers of these community shifts. We highlight the urgent need to prioritize conservation for declining species like *Dremomys pernyi* and *Mustela sibirica* and to enhance habitat connectivity to mitigate climate impacts. We recommend continuous monitoring of species interactions and environmental variables to safeguard this biodiversity hotspot. This study establishes a baseline for assessing ecosystem health in mountainous regions and demonstrates the value of integrating diversity metrics with network analysis for conservation planning.

## Figures and Tables

**Figure 1 biology-14-00897-f001:**
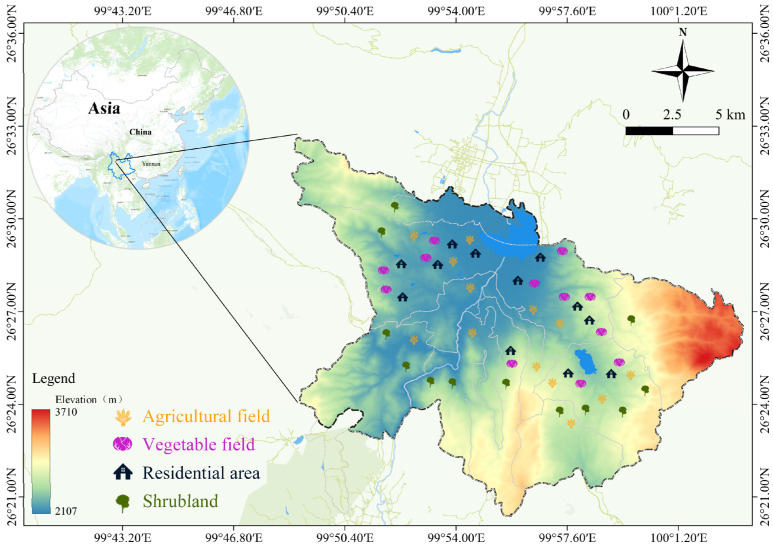
Study area in the Diannan area, Yunnan Province, China. The map in the upper left corner shows the geographic location of the Diannan area in China. The panel identifies the locations of the 48 sampling sites where trapping occurred from 2005 to 2017.

**Figure 2 biology-14-00897-f002:**
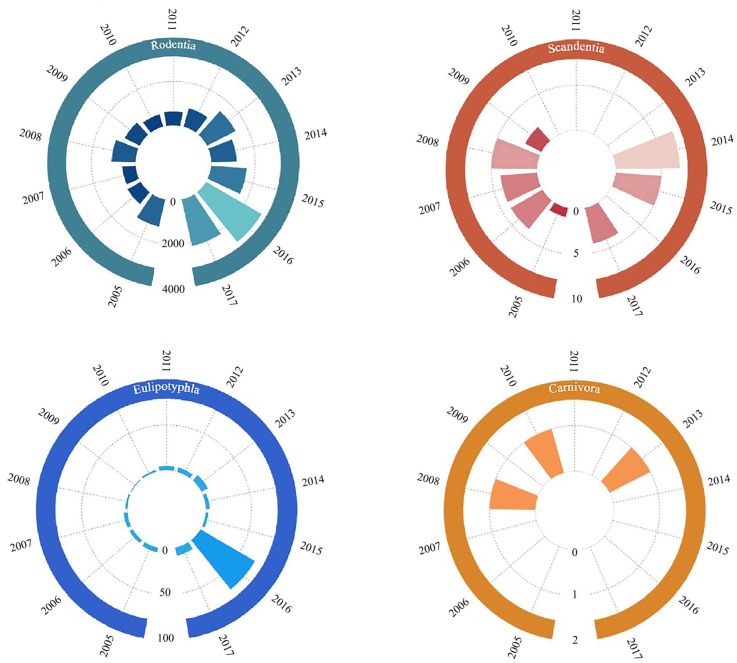
Chart of interannual variations in the numbers of small mammals in four orders of Diannan, Yunnan Province, China, 2005–2017. The green bars represent the number of individuals in the order Rodentia from 2005–2017; the red bars represent the number of individuals in the order Scandentia from 2005–2017; the blue bars represent the number of individuals in the order Eulipotyphla from 2005–2017; and the orange bars represent the number of individuals in the order Carnivora from 2005–2017.

**Figure 3 biology-14-00897-f003:**
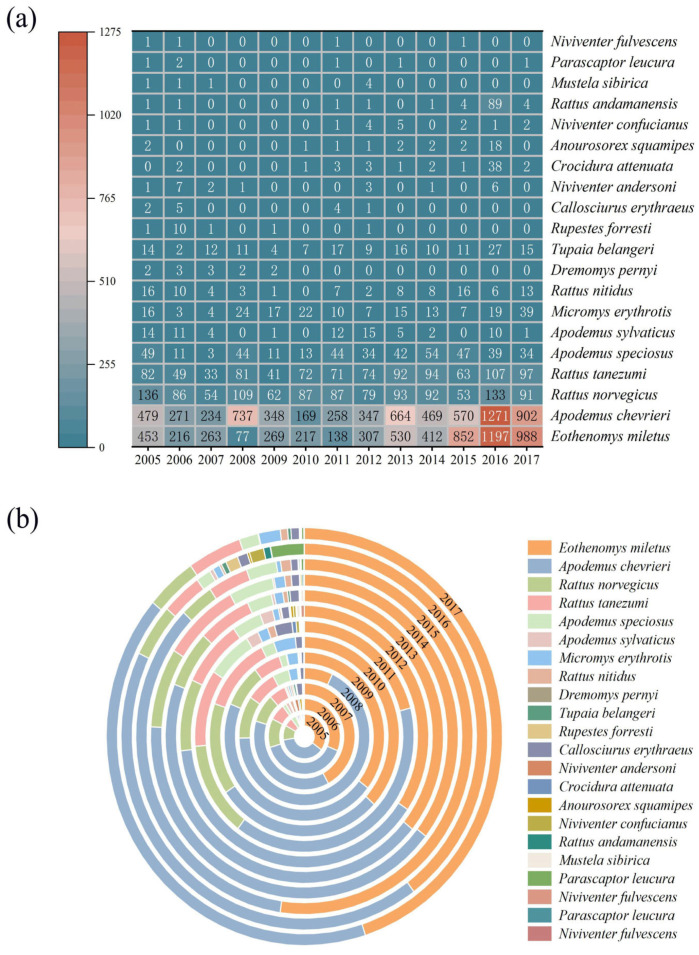
(**a**) Variation of small mammal species abundance in Diannan, Yunnan Province, China, 2005–2017 and (**b**) changes in relative abundance (%) of different small mammals in Diannan, Yunnan Province, China, 2005–2017.

**Figure 4 biology-14-00897-f004:**
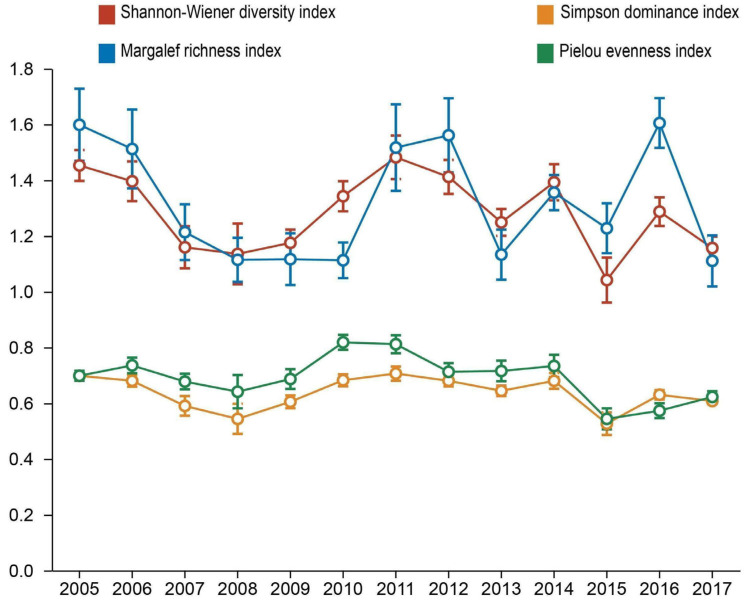
Changes in species diversity of small mammals of Diannan, Yunnan Province, China, 2005–2017. Points represent annual mean values of each diversity index, with vertical error bars indicating standard error (n = 48 sampling sites per year).

**Figure 5 biology-14-00897-f005:**
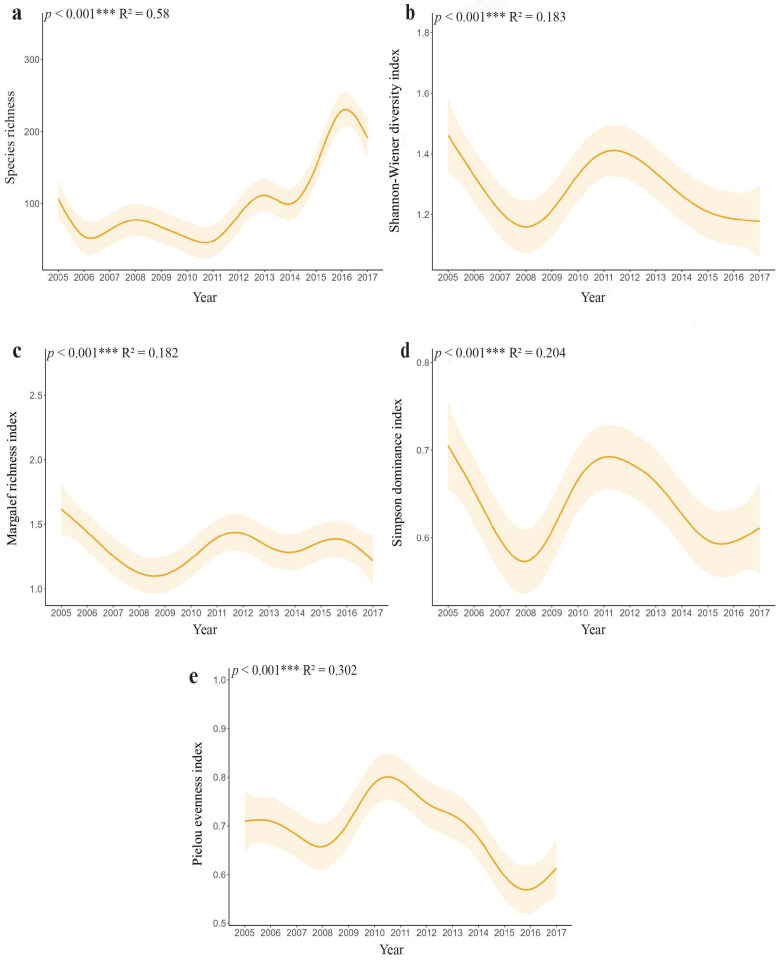
Generalized additive model (GAM) showing temporal trends in (**a**) species richness; (**b**) Shannon–Wiener index (*H*′); (**c**) Pielou’s evenness index (*J*′); (**d**) Simpson’s dominance index (*D*); and (**e**) Margalef’s richness index (*D_m_*) of small mammals in Diannan, Yunnan Province, China, 2005–2017. Shaded areas represent 95% confidence intervals. R^2^ quantifies the proportion of variation explained by time, while the *p* verifies trend significance (*** *p* < 0.001).

**Figure 6 biology-14-00897-f006:**
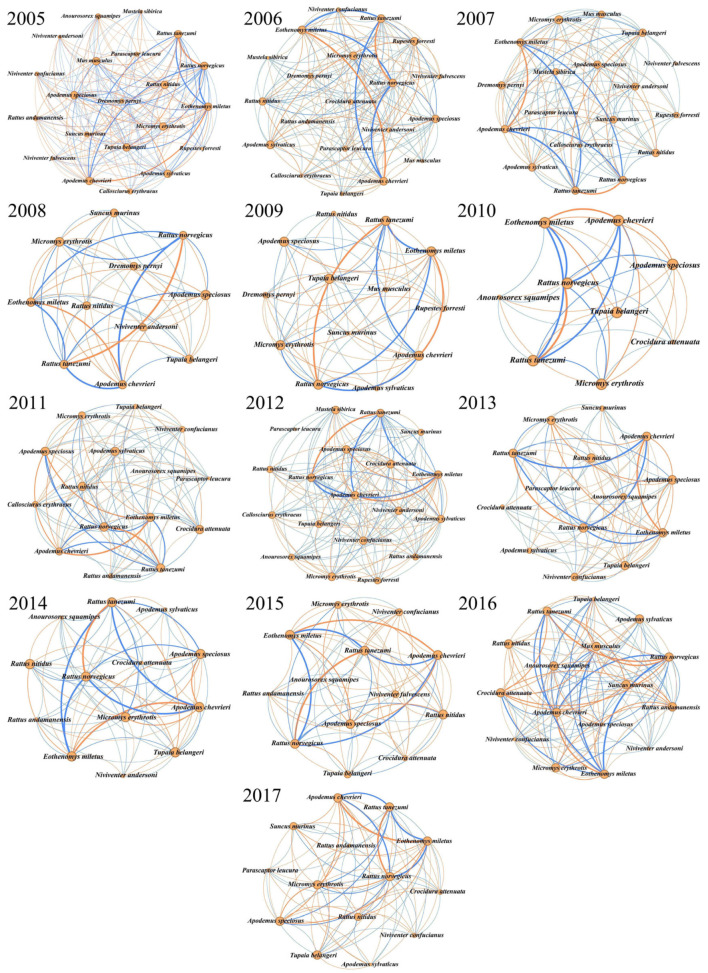
Small mammals’ co-occurrence networks in Diannan area from 2005 to 2017. Nodes represent species with size proportional to node degree. (The larger the node, the more links it has in the species co-occurrence network.) The orange links connecting the nodes represent positive associations. The blue links connecting the nodes represent negative associations.

**Figure 7 biology-14-00897-f007:**
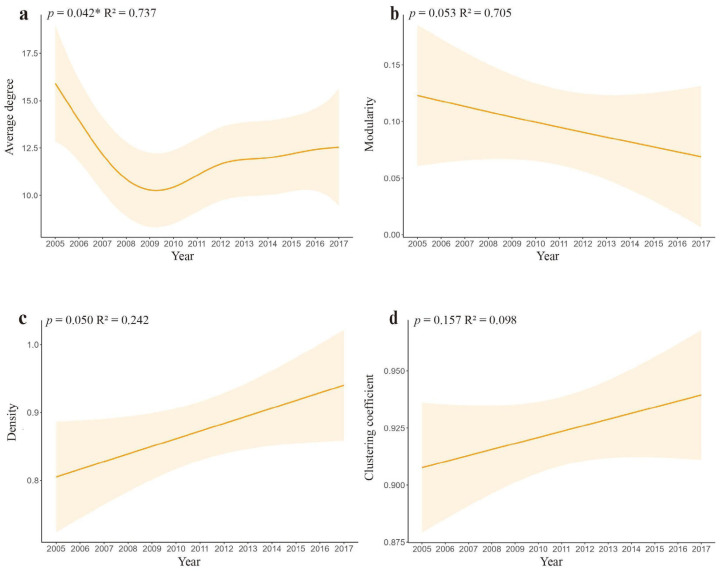
Generalized additive model (GAM) showing temporal trends of structural metrics of co-occurrence networks in Diannan, Yunnan Province, China, 2005–2017. (**a**) Average degree; (**b**) modularity; (**c**) density; and (**d**) clustering coefficient. Shaded areas represent 95% confidence intervals. R^2^ quantifies the proportion of variation explained by time, while the *p* verifies trend significance (* *p* < 0.05).

**Figure 8 biology-14-00897-f008:**
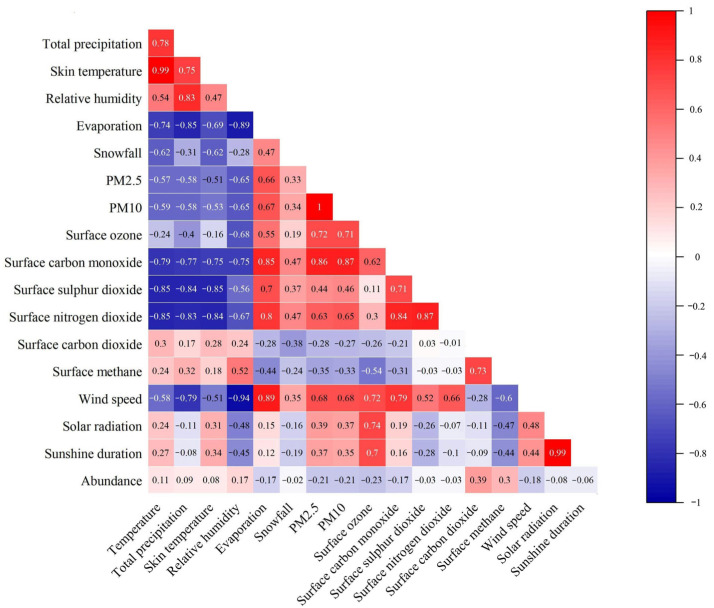
Pearson analysis of the abundance of small mammals and climatic factors in Diannan, Yunnan Province, China, 2005–2017. Color intensity represents the magnitude of correlation (red = positive, blue = negative).

**Figure 9 biology-14-00897-f009:**
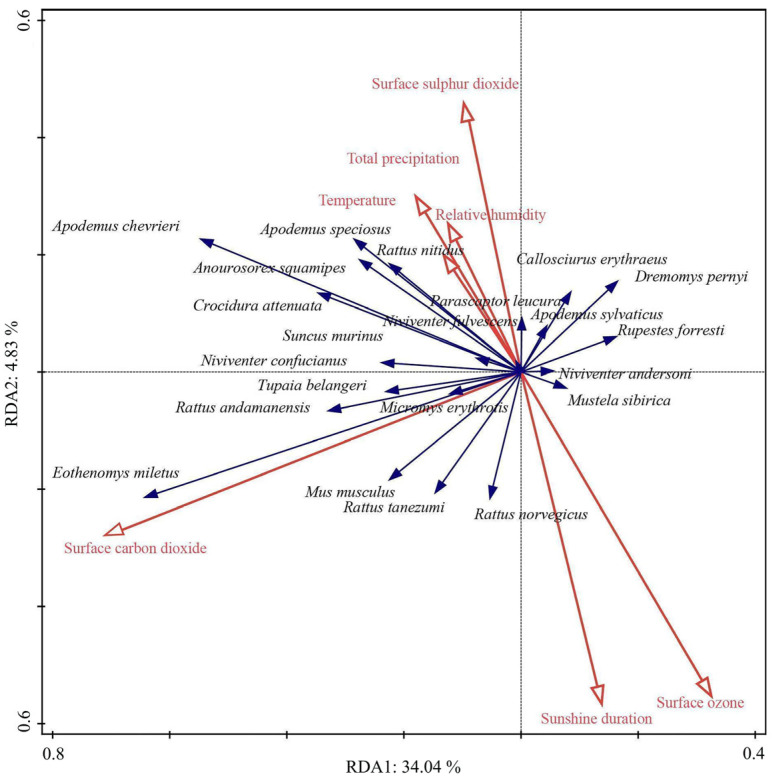
Redundancy analysis (RDA) of correlations between population size and climatic factors. Blue arrows represent small mammals, and red arrows represent climatic factors. The length of the climatic indicator arrow can represent the influence of the factors on the population size of small mammals. The angles between the arrows represent positive and negative correlations.

**Table 1 biology-14-00897-t001:** Species names and classification status of small mammals collected in the Diannan area, 2005–2017.

Order	Family	Genus	Species	Common Name
Rodentia	Cricetidae	*Eothenomys*	*Eothenomys miletus*	Large Chinese vole
	Muridae	*Apodemus*	*Apodemus chevrieri*	Chevrier’s field mouse
			*Apodemus speciosus*	Korean field mouse
			*Apodemus sylvaticus*	Wood mouse
		*Rattus*	*Rattus norvegicus*	Brown rat
			*Rattus tanezumi*	Oriental house rat
			*Rattus nitidus*	White-footed Indochinese rat
			*Rattus andamanensis*	Indochinese forest rat
		*Mus*	*Mus musculus*	House mouse
		*Micromys*	*Micromys erythrotis*	Red-eared harvest mouse
		*Niviventer*	*Niviventer confucianus*	Confucian niviventer
			*Niviventer andersoni*	Anderson’s niviventer
			*Niviventer fulvescens*	Chestnut white-bellied rat
	Sciuridae	*Dremomys*	*Dremomys pernyi*	Perny’s long-nosed squirrel
		*Callosciurus*	*Callosciurus erythraeus*	Pallas’s squirrel
		*Suncus*	*Suncus murinus*	Asian house shrew
Scandentia	Tupalldae	*Tupaia*	*Tupaia belangeri*	Northern tree shrew
Eulipotyphla	Talpidae	*Parascaptor*	*Parascaptor leucura*	White-tailed mole
	Soricidae	*Crocidura*	*Crocidura attenuata*	Asian gray white-toothed shrew
		*Suncus*	*Suncus murinus*	Asian house shrew
		*Anourosorex*	*Anourosorex squamipes*	Chinese mole shrew
Carnivora	Mustelidae	*Mustela*	*Mustela sibirica*	Siberian weasel

**Table 2 biology-14-00897-t002:** Indicators of co-occurrence network of small mammals in Diannan area in different years.

Years	Average Degree	Modularity	Density	Clustering Coefficient
2005	16.190	0.129	0.810	0.892
2006	15.900	0.226	0.837	0.891
2007	11.000	0.120	0.647	0.918
2008	9.455	0.015	0.945	0.945
2009	10.154	0.056	0.846	0.900
2010	7.778	0.019	0.972	0.972
2011	12.667	0.131	0.905	0.919
2012	14.632	0.151	0.813	0.914
2013	11.714	0.099	0.901	0.928
2014	10.462	0.020	0.872	0.907
2015	11.077	0.106	0.923	0.930
2016	14.750	0.098	0.983	0.983
2017	11.571	0.070	0.890	0.907

**Table 3 biology-14-00897-t003:** Changes in pairwise species co-occurrence connections between years.

	Stable		Lost		Gained	
	n	%Links	n	%Links	n	%Links
2005→2006	122	71.76	48	28.24	37	23.27
2006→2007	81	50.94	78	49.06	18	18.19
2007→2008	52	52.52	47	47.48	0	0
2008→2009	41	78.85	11	21.15	25	37.88
2009→2010	21	31.82	45	68.18	14	40.00
2010→2011	33	94.29	2	5.71	62	65.26
2011→2012	83	87.37	12	12.63	56	40.29
2012→2013	71	51.08	68	48.92	11	13.41
2013→2014	52	63.41	30	36.59	16	23.53
2014→2015	51	75.00	17	25.00	21	29.17
2015→2016	60	83.33	12	16.67	58	49.15
2016→2017	71	60.17	47	39.83	10	12.35

## Data Availability

The data that support the findings of this study are openly available in the Dryad Data Repository at https://datadryad.org/stash/share/4tjurMOfLCJubldbEQCyEfstEvegAcr5W3iKO-DRW6c (accessed on 20 July 2024).

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
