# Peer review of "Long-Term Monitoring Reveals Changes in the Small Mammal Community Composition and Co-Occurrence Patterns in the Diannan Area of Yunnan, China"

_biology, 2025, doi:10.3390/biology14070897_

Round 1

Reviewer 1 Report

Comments and Suggestions for Authors

The paper entitled "Long-term monitoring has revealed changes in the small mammal community composition and co-occurrence patterns in Diannan area in Yunnan, China" analyzes various aspects of small mammal populations in the Diannan area over a long period (17 years). 
The paper is clear in all its sections, providing a current context for the study in the introduction, clearly detailing the methods and results obtained, and discussing the results appropriately. The results are also illustrated graphically in an aesthetic and easy-to-read manner. 
I recommend a few minor improvements here:
-the statements in lines 50-51 should be cited
-as should those in line 62
-in the table header in line 82 for common names, I recommend using the term "common name" instead of "normal name"
-the explanation for Figure 6 is unclear, as both the orange and blue lines show a positive association
In line 168, the GAM models need to be cited
- in line 249, the title of chapter 3.5 refers to something other than the content of the chapter

Author Response

Reviewer: 1

Comments to the Author:

The paper entitled "Long-term monitoring has revealed changes in the small mammal community composition and co-occurrence patterns in Diannan area in Yunnan, China" analyzes various aspects of small mammal populations in the Diannan area over a long period (17 years). 

The paper is clear in all its sections, providing a current context for the study in the introduction, clearly detailing the methods and results obtained, and discussing the results appropriately. The results are also illustrated graphically in an aesthetic and easy-to-read manner. 

Minor comments:

I recommend a few minor improvements here:

Q1: The statements in lines 50-51 should be cited.

Response: We greatly appreciate your carefully considered comments, and we have added relevant citations from [Yang, C et al. Human expansion into Asian highlands in the 21st Century and its effects. Nat. Commun. 2022, 13, 4955. ] and [Yang, L.Yet al. Identifying the impact of global human activities expansion on natural habitats. J. Clean. Prod. 2024, 434, 140247.] and [Wei, G et al. How does rapid urban construction land expansion affect the spatial inequalities of ecosystem health in China? Evidence from the country, economic regions and urban agglomerations. Environ. Impact Assess. Rev. 2024, 106, 107533.] and [Xiong, S.W et al. Multiscale exploration of spatiotemporal dynamics in China's largest urban agglomeration: An interactive coupling perspective on human activity intensity and ecosystem health. J. Environ. Manag. 2025, 376, 124375.] to support these claims. (L56-57, page3).

Q2: As should those in line 62

Response: We greatly appreciate your carefully considered comments. Based on this suggestion, added relevant citations from [Cook, R.Net al. Environmental correlates of richness, community composition, and functional traits of terrestrial birds and mammals in a fragmented tropical landscape. Landsc. Ecol. 2020, 35, 2825-2841.] and [Cárdenas, P.A et al. Declines in rodent abundance and diversity track regional climate variability in North American drylands. Glob. Change Biol. 2021, 27, 4005-4023.] to support these claims(L68, page3).

Q3: In the table header in line 82 for common names, I recommend using the term "common name" instead of "normal name"

Response: Thank you for your kind suggestion. "Normal name" has been replaced with "Common name" as suggested.

Q4: In line 168, the GAM models need to be cited

Response: We appreciate your valuable comments. Based on this suggestion, added relevant citations from [Wood, S.N. Stable and efficient multiple smoothing parameter estimation for generalized additive models. J. Am. Statist. Ass. 2004, 99, 673-686.] and [Wood, S.N.; Goude, Y.; Shaw, S. Generalized additive models for large data sets. J. R. Stat. Soc. Ser. C-Appl. Stat. 2015, 64, 139-155.] to support these claims(L187, page7).

Q5: In line 249, the title of chapter 3.5 refers to something other than the content of the chapter

Response: We appreciate the constructive comments you .We have adjusted the title of chapter 3.5.

We sincerely appreciate your insightful comments and constructive suggestions on our manuscript.

Thank you once again for your time and consideration.

Reviewer 2 Report

Comments and Suggestions for Authors

General comments: 

The manuscript is well-structured and functions as a significant resource for both the global audience and the scientific community. Below, I have made an effort to emphasize several comprehensive review reports pertaining to the manuscript:

  1. The author(s) endeavored to explore the long-term monitoring that uncovers alterations in the composition of small mammal communities and their co-occurrence patterns in the Diannan region of Yunnan, China. However, it would also be advantageous to illustrate to the readers the importance of monitoring small mammals for the purposes of biodiversity conservation and ecosystem services, as well as to adjust your topic in accordance with this feedback.
  2. I believe that the topic is both innovative and relevant to the field. Additionally, it is expected to fill a void in the discipline.
  3. Although earlier research has focused on rodent and insectivore populations in this region, there is still a deficiency in systematic knowledge concerning the composition of small mammal communities, their diversity dynamics, and their interactions with environmental variables. Therefore, this study undertook the first comprehensive analysis utilizing 13 years of continuous monitoring data.
  4. In my opinion, the research utilized all essential methodological approaches and analytical instruments to reveal important findings for this study. Nevertheless, it would be beneficial to incorporate the total area coverage, faunal species, and the current anthropogenic activities present in the region. Additionally, it is crucial to examine how these factors are affecting biodiversity and ecosystem services within the context of the study area description. Furthermore, there are several questions regarding clarity and requests for revisions that need to be addressed.
  5. The conclusion provided is consistent with the findings and responds to the main research question. The research produced a sound conclusion, and the study is supported by sufficient data.
  6. Almost all of the reference lists are appropriate and up-to-date. However, there are a small number of outdated sources that ought to be replaced with more recent research.
  7. All tables and figures are displayed appropriately and in a suitable manner.
  8. I offered multiple editorial and clarity recommendations concerning the document that was reviewed.

Comments on the Quality of English Language

I have offered multiple editorial and clarity recommendations concerning the document that was reviewed. Kindly enhance the document in accordance with these suggestions.

Author Response

Reviewer: 2 

Comments to the Author:

General comments: 

The manuscript is well-structured and functions as a significant resource for both the global audience and the scientific community. Below, I have made an effort to emphasize several comprehensive review reports pertaining to the manuscript:

  1. The author(s) endeavored to explore the long-term monitoring that uncovers alterations in the composition of small mammal communities and their co-occurrence patterns in the Diannan region of Yunnan, China. However, it would also be advantageous to illustrate to the readers the importance of monitoring small mammals for the purposes of biodiversity conservation and ecosystem services, as well as to adjust your topic in accordance with this feedback.

Response: We have made correction according to the Reviewer’s comments.

2.I believe that the topic is both innovative and relevant to the field. Additionally, it is expected to fill a void in the discipline.

Response: Thanks.

3.Although earlier research has focused on rodent and insectivore populations in this region, there is still a deficiency in systematic knowledge concerning the composition of small mammal communities, their diversity dynamics, and their interactions with environmental variables. Therefore, this study undertook the first comprehensive analysis utilizing 13 years of continuous monitoring data.

Response: Thanks.

4.In my opinion, the research utilized all essential methodological approaches and analytical instruments to reveal important findings for this study. Nevertheless, it would be beneficial to incorporate the total area coverage, faunal species, and the current anthropogenic activities present in the region. Additionally, it is crucial to examine how these factors are affecting biodiversity and ecosystem services within the context of the study area description. Furthermore, there are several questions regarding clarity and requests for revisions that need to be addressed.

Response: We have made correction according to the Reviewer’s comments.

5.The conclusion provided is consistent with the findings and responds to the main research question. The research produced a sound conclusion, and the study is supported by sufficient data.

Response: Thanks.

6.Almost all of the reference lists are appropriate and up-to-date. However, there are a small number of outdated sources that ought to be replaced with more recent research.

Response: We have made correction according to the Reviewer’s comments.

7.All tables and figures are displayed appropriately and in a suitable manner.

Response: Thanks.

8.I offered multiple editorial and clarity recommendations concerning the document that was reviewed.

Response: We have made correction according to the Reviewer’s comments.

Comments on the Quality of English Language:I have offered multiple editorial and clarity recommendations concerning the document that was reviewed. Kindly enhance the document in accordance with these suggestions.

Response: We have made correction according to the Reviewer’s comments.

We sincerely appreciate your insightful comments and constructive suggestions on our manuscript. We have carefully addressed all the points raised by incorporating modifications throughout the manuscript. Below is a detailed response to each comment:

  1. Conservation significance:We have enhanced the introduction and abstract to emphasize the importance of small mammal monitoring for biodiversity conservation and ecosystem services, citing recent studies (Balčiauskas, L.; Benedek, M.A. Advances in diversity and conservation of terrestrial small mammals. Div2023, 15, 884.).

  1. Study area details:Comprehensive information about the total area coverage (244km²), The specific location of the survey area is also explained in the survey methods section. We have supplemented information on the main plants in this area. As for the fauna, since no researchers have conducted a separate survey in this region before, we do not have complete fauna information, and we sincerely apologize for this.

  1. Reference updates:Someoutdated references have been replaced with current literature .

  1. Language edits: All editorial suggestions regarding English clarity and grammar have been implemented, including:

   - Tense consistency corrections

   - Simplified complex sentences in Results

   - Improved word choice throughout

  1. Structural Improvements:

- The Simple Summary now better highlights the ecological significance of the study.

- The Abstract has been refined to more clearly present key findings.

- The Conclusion section has been strengthened to better reflect the study's contributions to the field.

  1. Statistical Clarification: Added detailed description of the statistical methods employed. Included specific p-values and effect sizes where appropriate.

  1. Formatting Standardization: We have carefully addressed all issues related to figures and tables presentation.

All modifications have been tracked in the revised manuscript for your convenience. We believe these changes have significantly improved the quality and clarity of our work, and we thank the reviewers for their valuable input.

Thank you once again for your time and consideration.